# Single-photon detection and cryogenic reconfigurability in lithium niobate nanophotonic circuits

Emma Lomonte[1,2,3], Martin A. Wolff [1,2,3], Fabian Beutel[1,2,3], Simone Ferrari[1,2,3], Carsten Schuck [1,2,3], Wolfram H. P. Pernice [1,2,3 ✉] & Francesco Lenzini [1,2,3 ✉]

Lithium-Niobate-On-Insulator (LNOI) is emerging as a promising platform for integrated quantum photonic technologies because of its high second-order nonlinearity and compact waveguide footprint. Importantly, LNOI allows for creating electro-optically reconfigurable circuits, which can be efficiently operated at cryogenic temperature. Their integration with superconducting nanowire single-photon detectors (SNSPDs) paves the way for realizing scalable photonic devices for active manipulation and detection of quantum states of light. Here we demonstrate integration of these two key components in a low loss (0.2 dB/cm) LNOI waveguide network. As an experimental showcase of our technology, we demonstrate the combined operation of an electrically tunable Mach-Zehnder interferometer and two waveguide-integrated SNSPDs at its outputs. We show static reconfigurability of our system with a bias-drift-free operation over a time of 12 hours, as well as high-speed modulation at a frequency up to 1 GHz. Our results provide blueprints for implementing complex quantum photonic devices on the LNOI platform.

[1] Institute of Physics, University of Muenster, 48149 Muenster, Germany. [2] CeNTech—Center for Nanotechnology, 48149 Muenster, Germany. [3] SoN—Center for Soft Nanoscience, 48149 Muenster, Germany. ✉email: wolfram.pernice@uni-muenster.de; lenzini@uni-muenster.de

On-chip integration of single-photon detectors and reconfigurable optical circuits is a crucial step toward a fully scalable approach to quantum photonic technologies[1]. By confining light inside lithographically patterned waveguides, single photons can be actively routed[2–4] and interfered[5–7] in miniaturized reconfigurable optical networks, and their state can be read out with on-chip detectors[8,9]. Integration of these two key elements on a common platform enhances the scalability of quantum photonic devices by minimizing their footprint and eliminating the need for lossy interconnects between separated optical systems.

Superconducting nanowire single-photon detectors (SNSPDs) are currently the technology of choice for single-photon detection in waveguides, featuring an on-chip detection efficiency in excess of 90%[8,10]. Besides their high efficiency and potential for scalability, waveguide-integrated SNSPDs have shown superior performance overall their competing platforms, including negligible dark counts[11] and the possibility of achieving GHz count rates at a low timing jitter[12,13]. A key question to date is the choice of an optimal material platform where SNSPDs and reconfigurable circuits can coexist and be efficiently operated in a cryogenic environment.

Reconfigurable optical elements, such as electrically tunable phase shifters, are implemented in Silicon waveguides or other conventional photonic platforms by means of either free-carrier injection[14,15] or thermo-optical tuning[16,17]. While the former approach is intrinsically lossy and thus not suitable for quantum photonic applications, the latter comes with the disadvantage of thermal crosstalk and large power dissipation. Furthermore, the thermo-optic coefficient of silicon and other common nanophotonic material systems decreases by several orders of magnitude at cryogenic temperature[18,19], making the use of thermo-optical devices not possible or highly inefficient in combination with SNSPDs. An alternative tuning mechanism compatible with cryogenic operation can be provided by the use of microelectromechanical systems[20], and their integration with on-chip detectors has been recently demonstrated on the silicon nitride platform[21]. Although these devices can now reach acceptable performance, including insertion loss <0.5 dB and a ~2 V driving voltage with a compact footprint[22], they require a complex procedure for the fabrication of free-standing optical structures and are prone to mechanical damage, making their use challenging for the realization of large-scale photonic integrated circuits. Moreover, optomechanical devices suffer from a maximum modulation bandwidth limited to the ~MHz range, which prevents their usage for a large number of important applications—such as quantum computing protocols based on time-bin encoding[23,24], spatial- and time-multiplexing schemes for scalable quantum computing[2,25,26], or fast feedforward operations for measurement-based quantum computation[27–29] – where a bandwidth in the range of few hundreds of MHz up to the ~GHz regime is mandatory.

Electro-optic modulators (EOMs) based on the Pockels effect can overcome all the aforementioned limitations, and provide a simple and cryogenic-compatible[30–33] platform for on-chip reconfigurable photonics. In this context, thin Lithium Niobate films bonded onto a silica insulating substrate (LNOI: lithium-niobate-on-insulator) have recently emerged as a particularly attractive technology for the realization of waveguides with submicron scale in $\chi^{(2)}$-nonlinear materials[34]. Waveguides patterned by electron-beam lithography on thin-film LN have demonstrated ultra-low propagation loss in both the visible[35] (6 dB/m) and telecom[36] (2.7 dB/m) wavelength ranges, and already enabled the realization of EOMs with a bandwidth up to ~100 GHz[37]. Furthermore, owing to strong light confinement and compatibility with periodic poling techniques, LNOI waveguides have enabled unprecedented levels of efficiency in second-order

nonlinear frequency conversion processes[38,39]. Thus, LNOI also holds great promise for implementing all the required operations for quantum photonic technologies—single-photon detection, quantum state manipulation, and nonlinear state generation—on a single monolithic platform.

With the recent experimental demonstration of SNSPDs patterned on top of LNOI waveguides[40], co-integration with electro-optically tunable LNOI circuits operating at cryogenic temperature remains an outstanding challenge for realizing a programmable high-speed quantum photonic processor. Here, we report the realization of SNSPDs and EOMs monolithically integrated in a reconfigurable LNOI waveguide network. We assess the performance of our detectors by measuring their detection efficiency, dead time, and timing jitter, and demonstrate the combined operation of SNSPDs and an EOM by performing electro-optic switching between two detectors fabricated at the two outputs of a Mach–Zehnder interferometer. We show static reconfigurability of our system with a bias-drift-free operation over a time of 12 h, as well as high-speed modulation at a frequency up to 1 GHz.

## Results and discussion

**Configuration of the integrated device.** Figure 1a, b shows a microscope image of the integrated device employed for demonstrating the joint cryogenic operation of SNSPDs and an EOM, and a schematic of the experimental setup, respectively. The device consists of a balanced and tunable Mach–Zehnder interferometer (MZI) made of an electro-optic phase shifter and two waveguide-integrated SNSPDs at the two outputs. The photonic chip is placed inside a closed-cycle He-4 cryostat operated at a base temperature $T \simeq 1.3$ K and mounted on a three-axis piezoelectric stage. A fiber v-groove array with a pitch of 127 μm and an RF contact probe with a pitch of 125 μm (see Fig. 1b) is used to couple light into and out of the optical circuit, and to drive the electronic circuits of the EOM and the two SNSPDs, respectively.

Optical circuits (see Methods for details on the fabrication process) are implemented as ridge waveguides fabricated by electron-beam lithography and Argon milling on a 300 nm thick X-cut LN film bonded on a Silica-on-Silicon wafer. The waveguides are clad with a 750 nm thick hydrogen silsesquioxane (HSQ) layer and designed with a top width equal to 1.1 μm to ensure single-mode operation for TE-polarized light in the 1550 nm wavelength range. Coupling of the light from single-mode optical fibers to the waveguides is achieved by the use of four apodized surface grating couplers with a negative diffraction angle and a peak coupling efficiency $\simeq -4.5$ dB[41].

Attenuated CW laser light from a tunable telecom laser enters the circuit either from In1 or In2 and is evenly split into the two arms of the MZI by a first directional coupler (DC). The two optical paths of the interferometer and three 1.7 mm long gold electrodes in a Ground-Signal-Ground configuration are patterned along the Y-axis of the crystal to provide an efficient overlap between the fundamental TE mode of the waveguide and the Z-component of the applied electric field via the highest electro-optic component ($r_{33} \simeq 30$ pm/V) of the LN susceptibility tensor (see Fig. 1c). Unlike the previously reported implementations of EOMs in LNOI circuits[37,42], where signal and ground electrodes were placed at the two sides of the waveguides, here we opted to pattern our electrodes on top of the employed HSQ cladding. This choice was made to allow for the low-loss crossing of the electrodes with the waveguides without any need for performing complex additional fabrication steps[42]. Light at the output of the second DC of the MZI is guided toward two waveguide-integrated SNSPDs, labeled Det1 and Det2 in Fig. 1a. Before reaching the single-photon detectors, two additional DCs

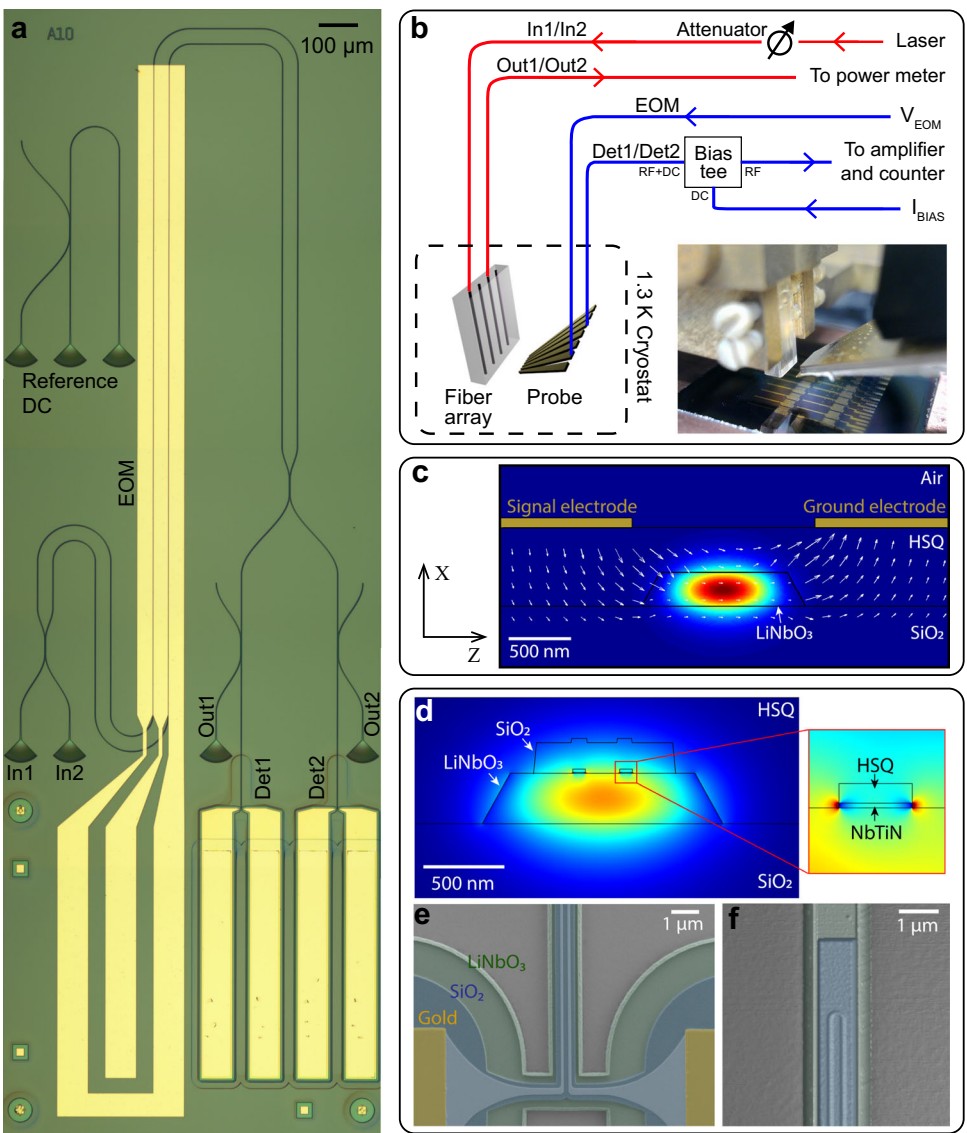

**Fig. 1 Configuration of the chip and the experimental setup. a** Optical microscope image of the integrated device. The reference directional coupler (DC) at the upper left of the figure allows for measuring the splitting ratio of the DCs upstream from the detectors, and for estimating the insertion loss of the MZI. **b** Schematic of the experimental setup with optical (red) and electrical (blue) access to the LN-chip. Inset in the figure shows a photograph of the photonic chip, the fiber array, and the contact probe employed in our experiment. **c** Electrode configuration used for the realization of the electro-optic phase shifter. The colormap is the field intensity of the fundamental TE mode supported by the waveguide calculated with a finite-difference mode solver. White arrows are the lines of the electric field applied to the waveguide calculated with a finite-difference mode solver. **d** Schematic of the nanowire configuration used for the realization of waveguide-integrated SNSPDs, and simulation of the nanowire absorption. The colormap is the field amplitude of the fundamental TE mode supported by the waveguide as calculated with a finite-element model solver. **e** False-color scanning electron micrograph of a waveguide-integrated SNSPD taken in the proximity of the gold electrodes pads. **f** Surface topography of a waveguide-integrated SNSPD taken in the proximity of the U-turn by atomic force microscopy. The same color scheme of Fig. 1e is here used for clarity. Both images were taken after waveguide etching, before cladding the device with HSQ.

separate a fraction of the light into two observation paths of equal intensity so that it can also be coupled out from Out1/Out2. These two reference ports can be used to estimate the on-chip detection efficiency of the SNSPDs by measuring the output light with a power meter (see Methods), or to test the performance of the EOM with two fast photodiodes.

SNSPDs (see Fig. 1d–f) are formed by patterning two U-shaped niobium titanium nitride (NbTiN) nanowires with a thickness of ~5 nm and a width of 75 nm on top of the waveguides, whereby a 200 nm thick RF-sputtered $SiO_2$ cover layer is applied to protect the nanowires during our top-down multi-step fabrication process (see Methods for further details). For the configuration depicted in Fig. 1d, we numerically estimate an absorption rate of our nanowires $\simeq 0.35$ dB/μm, leading to an overall absorption >30 dB for the chosen detector length of 100 μm. To bias the detectors, two independent voltage sources are connected in series with a 1 MΩ resistor to deliver a stable bias current of around $\simeq 10$ μA. To read out the weak RF detector signal from the nanowires, low noise amplifiers are connected at the outputs of the detectors, and a time tagger is employed for recording the detector's counts. Since the DC bias voltage and the RF detection signal of the nanowires propagate through the same line in the cryostat, a bias-tee is inserted into the electronic circuit to decouple the DC bias from the RF response signal (see Fig. 1b).

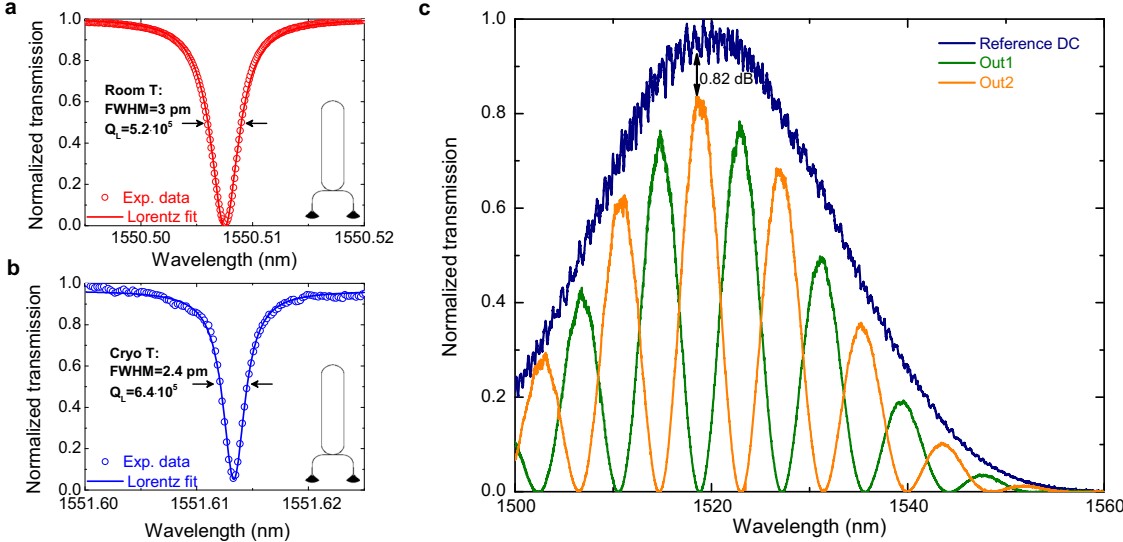

**Fig. 2 Characterization of the device loss. a** Resonance of a racetrack resonator (fabricated on the same chip of the device in Fig. 1) in the critical coupling regime measured at room temperature. The extracted loaded $Q$ factor is $Q_L = 5.2 \times 10^5$ (intrinsic $Q \simeq 10^6$). The inset in the graph shows a sketch of the racetrack cavity used to determine the loss. A bus waveguide connected to two grating couplers is used to couple light into the resonator. **b** Resonance of the same racetrack resonator measured at cryogenic temperature. The extracted loaded $Q$ factor is $Q_L = 6.4 \times 10^5$ (intrinsic $Q \simeq 1.2 \times 10^6$). **c** Transmission spectrum of the reference DC device at the upper left of Fig. 1a, and of the two outputs of the MZI measured when laser light is injected into In2. The insertion loss of the MZI is estimated by fitting the transmission spectrum of the reference DC with a Gaussian function, the spectrum of Out1/Out2 with a Gaussian function multiplied by a sinusoidal function, and by comparing the maxima of the two Gaussians. The insertion loss averaged over six identical devices fabricated on the same chip is found equal to $(0.82 \pm 0.24)$ dB.

For Det2 we make use of a cryogenic-temperature amplifier to reduce the electrical noise of the amplified signal and assess the timing jitter of our nanowires (see the section below).

The propagation loss of our waveguides was characterized both at room- and cryogenic- temperature by measuring the Q-factor of the ring- and racetrack-resonators fabricated on the same chip (see Fig. 2a, b and Methods for further details). At room temperature, we determined propagation loss equal to 0.22 dB/cm in a straight waveguide and to 0.68 dB/cm in the case of circular bends with a 70 μm radius. At cryogenic temperature, we found comparable propagation loss equal to 0.20 dB/cm in a straight waveguide and to 0.60 dB/cm for a circular bend. While at room temperature critical coupling was achieved, at a cryogenic temperature a slightly reduced extinction ratio of the measured resonances introduced an estimated error of around ±0.05 dB/cm in the inference of the propagation loss. We note that the extracted value of propagation loss is close to the *status quo* for single-mode LNOI waveguides (waveguide width $\simeq 1$ μm)[36]. Insertion loss of the MZI—which includes propagation loss in the straight LN waveguides, metal-induced absorption loss, insertion loss of two DCs, and three electrode-waveguide crossings—was estimated by comparing the device transmission with that of the reference DC at the upper left of Fig. 1a, and found equal to $\simeq 0.8$ dB (see Fig. 2c).

### Performance of waveguide-integrated SNSPDs.
Only recently, the first SNSPDs integrated with LNOI waveguides were reported, using a ~5 nm niobium nitride (NbN) thin film as a super-conducting material[40]. Although an on-chip detection efficiency (OCDE) $\simeq 46\%$ was achieved, the waveguides showed elevated levels of propagation loss (5 dB/cm) due to the employed process for fabrication of superconducting nanowires based on a bottom-up approach. In this work, we use a top-down fabrication for LNOI waveguide-integrated SNSPDs and achieve substantially lower propagation loss (0.2 dB/cm). For our experimental implementation, the use of niobium–titanium nitride (NbTiN)

nanowires was chosen over NbN because of their low kinetic inductance[43,44], which enables high-speed detection, and low dark count rate[11]. Its polycrystalline phase also eases the growth on several substrates, granting high uniformity and high yield production even for very thin films[45].

To assess the OCDE, CW laser light was coupled into the circuit and attenuated to a calibrated photon flux $\Phi = 10^6$ photons/s at the input of the SNSPDs (see Methods for further details). From the measured detector count rates, we calculate the OCDEs of the two SNSPDs at the outputs of the MZI, which are shown in Fig. 3a, b as a function of the applied bias current. For Det1 we measured a critical current $I_c$ of 14 μA and achieved a maximum OCDE $\simeq 24\%$. A similar maximum OCDE $\simeq 27\%$ was found for Det2, with a slightly lower critical current of 12.5 μA. We attribute this imperfect detection efficiency to either an incomplete etching of the superconducting film in the detector region, or to the redeposition of sputtered material on the waveguide sidewalls during Ar etching of the LN film which is not removed in the proximity of the nanowires (see the Fabrication section in Methods, and Supplementary Note 1 for more details). Despite this, the presence of a plateau of nearly saturating detection efficiency for increasing bias currents (see Fig. 3a) indicates high internal quantum efficiency of our detectors[46] and the possibility of increasing the OCDE with further improvements to the fabrication process. Dark count rates (not shown in the figures) were characterized by biasing the detectors at 85% of their critical current and integrating over a time of 20 min and found in the range of 2 cps (counts-per-second) for both detectors.

Further important parameters for the characterization of SNSPDs are the dead time and the timing jitter of the detector. These are particularly important metrics for applications requiring high-speed electro-optic modulation. For example, in quantum computing protocols based on time-encoding the dead time of the detector sets the maximum achievable modulation rate and the required length of the optical delay lines[24], while the

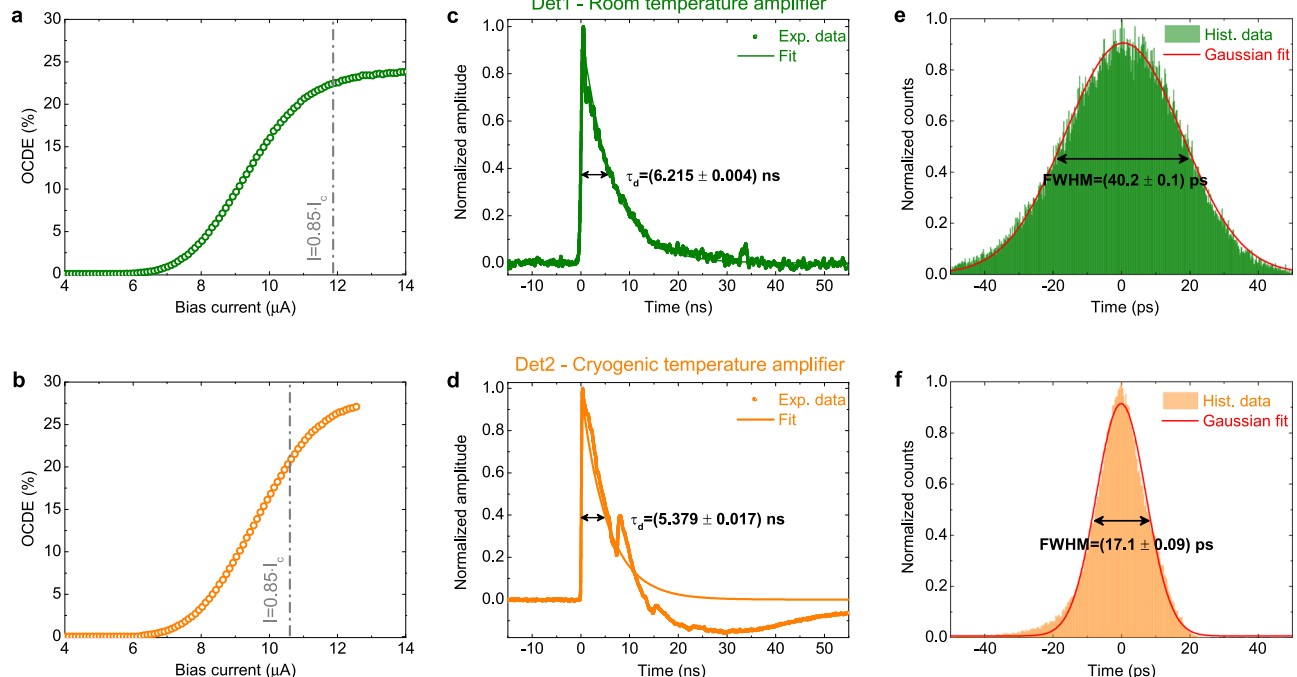

**Fig. 3 Performance of the waveguide-integrated detectors. a, b** Measured on-chip detection efficiency (OCDE) of the two detectors as a function of the bias current. The dashed-dotted line indicates the value of the current $I = 0.85 \times I_c$, at which the two detectors were biased to measure the timing jitter and to show the joint operation of the EOM and the SNSPDs. **c, d** Output electrical signals of the two detectors registered with a digital oscilloscope upon the absorption of a photon. **e, f** Time histograms of the photon counts used to measure the timing jitter of the two detectors (see main text for an explanation). Det2 is connected to a cryogenic temperature amplifier, which allows to reduce the electrical noise of the amplified signal and measure a lower timing jitter.

jitter determines the uncertainty in the arrival time of a photon and ultimately the achievable detection timing resolution.

To assess the dead time of an SNSPD—defined as the time interval in which a detector is not able to produce any electrical response after a detection event—a relevant parameter is the decay time of the detector. This was estimated for the two SNSPDs (see Fig. 3c, d) by measuring their electric output trace with a digital oscilloscope. Upon absorption of a single photon, the nanowire enters a transient resistive state and a "nanowire click", represented by a spike in the measured voltage, is observed. The decay time of the detector is then defined as the time interval needed for the electrical signal to decrease to 1/e of its maximum value. This has been estimated to be in the range of ≃6 ns for both detectors, a value comparable to state-of-the-art SNSPDs with a similar geometry[45].

The timing jitter was determined by performing start-stop measurements with a pulsed laser source operating at a repetition rate of 40 MHz at a wavelength of 1550 nm. During the measurement, both detectors were biased at 85% of the critical current. By using the electrical SNSPD output signal as a start trigger and the electrical reference output of the laser as a stop signal, we record a time histogram, which is shown in Fig. 3e, f for the two detectors. We then calculate the jitter as the full width at half the maximum of the Gaussian function used to fit the histogram. Since the electrical jitter of room-temperature low-noise amplifiers makes a significant contribution to the overall jitter, to assess the timing performance of our nanowires we make use of a cryogenic-temperature amplifier for Det2. This allowed us to measure a timing jitter as small as ≃17 ps, which is competitive for waveguide-integrated SNSPDs[46].

**The combined operation of EOM and SNSPDs: low-speed and DC operation.** To test the combined operation of EOM and

SNSPDs, we made use of the setup already described in the previous section and depicted in Fig. 1b. Attenuated laser light was coupled into In2 to obtain a calibrated photon flux $\Phi = 10^6$ photons/s at the input of the detectors. The RF contact probe was used to deliver a driving voltage $V_{EOM}$ to the electro-optic phase shifter, and the modulated optical signals at the output of the MZI were directly monitored from the counts recorded with the two SNSPDs. For all the below measurements, the two detectors were biased at 85% of their critical current.

To evaluate the half-wave voltage ($V_\pi$) of our EOM, the electro-optic phase shifter was driven with a ramp function with a peak-to-peak amplitude of 20 V and a frequency of 1 kHz. The output trigger of the function generator used to drive the EOM was connected to the start channel of a time tagger, and the outputs of the two SNSPDs to the stop channels, effectively functioning as a single-photon sensitive oscilloscope for registering the modulation of weak optical signals. The resulting time histograms are reported in Fig. 4a for the two detectors. The two signals showed a sinusoidal oscillation over time with a maximum phase shift slightly larger than $\pi$ for the applied voltage of $20 V_{pp}$. By fitting the normalized count rates of the two detectors with a sinusoidal function we extracted a half-wave voltage at cryogenic temperature $V_\pi^{cryo} = 17.8$ V (see Fig. 4b). By comparison, the half-wave voltage measured at room temperature using two standard photodiodes and the same ramp function was found equal to $V_\pi^{RT} = 15.5$ V. We attribute this change in the half-wave voltage of the EOM to a slight reduction of the $r_{33}$ coefficient of LN at cryogenic temperature, an effect already observed in bulk LN crystals[31] as well as LN waveguides fabricated by Ti-indiffusion[32]. Given the length of the electrodes of our EOM (1.7 mm), we estimate voltage-length products $V_\pi \cdot L \simeq 2.6$ V cm at room temperature, and $V_\pi \cdot L \simeq 3$ V cm at cryogenic temperature.

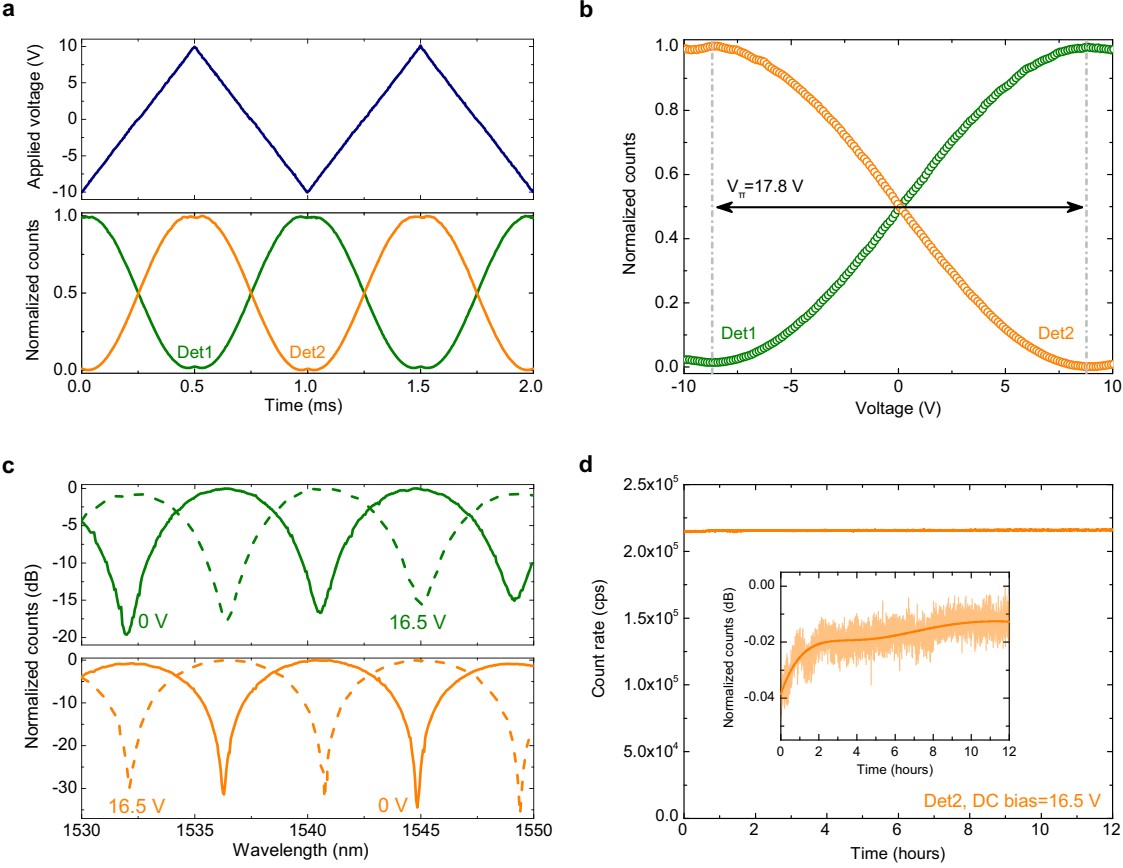

**Fig. 4 Combined operation of EOM and SNSPDs: slow-speed and DC operation. a** Normalized count rates collected from the two detectors with a time tagging module (lower image) when the EOM is driven with a ramp function with an amplitude of 20 $V_{pp}$ and a frequency of 1 kHz (upper image). **b** Zoom in on the detector counts showing a half-modulation period of the time histogram of Fig. 3a. The plotted data is used to estimate the half-wave voltage of the EOM. **c** Collected count rates from Det1 (upper image) and from Det2 (lower image) as a function of the laser wavelength for an applied DC bias of 0 V (solid line) and of 16.5 V (dashed line). Data are plotted on a dB scale. **d** Count rate collected from Det2 over a period of 12 h at an applied DC bias of 16.5 V. At each data point the photon counts are integrated over a time of 10 s. The inset in the figure shows the normalized count rate plotted in a dB scale. During the measurement, the laser wavelength was set to a value for which all the light entering the MZI is directed to Det1 for a zero applied voltage to the EOM. Upon the application of a DC bias equal to the $V_\pi$ of the EOM, the optical outputs of the MZI are inverted and all the input light directed to Det2.

When driving the EOM with a DC bias, a slightly smaller voltage $V_\pi = 16.5$ V allowed us to completely switch the outputs of the MZI between their on and off states. In Fig. 4c, we report the normalized counts collected from the two detectors as a function of the laser wavelength for the case of an applied bias voltage of 0 V (solid line) and of 16.5 V (dashed line). The data were obtained by tuning the laser wavelength at steps of 0.1 nm and integrating the counts over a time of 100 ms at each step. The count rates collected from Det2 displayed an extinction ratio between on and off states >30 dB, indicating excellent operation of our EOM and a high signal-to-noise ratio of our detectors even in presence of the applied DC bias. The lower extinction ratio ($\simeq$15 dB) observed for Det1 is due to an imperfect 50:50 splitting ratio of the two-directional couplers of the MZI, which we determined equal to $\simeq$56% from our measurements.

It is well-known that, when driven with a DC voltage, EOMs implemented on LN substrates suffer from a parasitic bias-drift, which must be overcome either by appropriate engineering of the material stack or by using complex feedback bias controllers to compensate for the shift of the EOM from its optimal operating point[47]. This problem has been recently reported also for LNOI waveguides, and the use of thermo-optic phase shifters proposed a solution for tuning the operating point of the EOM[48]. Here, to characterize the bias-drift of our device at cryogenic temperature, we recorded the photon counts from Det2 over a period of 12 h at

an applied DC bias of 16.5 V. The measured count rate is reported in Fig. 4d as a function of the acquisition time, and plotted in the inset in a dB scale. The recorded signal was found highly stable in time, with an overall variation of less than 0.05 dB over the measurement time window. Importantly, this result demonstrates the suitability of our platform for several applications in quantum photonics—e.g., a reconfigurable boson sampler[49,50] - where only static rather than fast reconfigurability is required.

A remarkably different behavior was observed at room temperature: our EOM could be effectively operated only with an AC signal and, when applying a DC bias, the optical outputs of the MZI quickly returned to their original state for a zero applied voltage on a time scale of less than one second. We argue that this is due to the presence of a charge flow in either the LN crystal or the HSQ cladding, and redistribution of electric charges on the waveguide sidewalls counteracting the applied electric field. At present, it is not possible to identify a precise reason for the enhanced stability of our EOM at cryogenic temperature. Indeed, DC-bias-drift is regarded as a complex phenomenon whose source can be attributed to a large number of possible effects, including the build-up of pyroelectric charges, crystal defects introduced during fabrication, or ionic contamination of the buffer layer[47]. We note that similar behavior has been reported for LN waveguides fabricated by Ti-indiffusion, where the temperature dependence of the drift rate was found to follow an Arrhenius distribution[51].

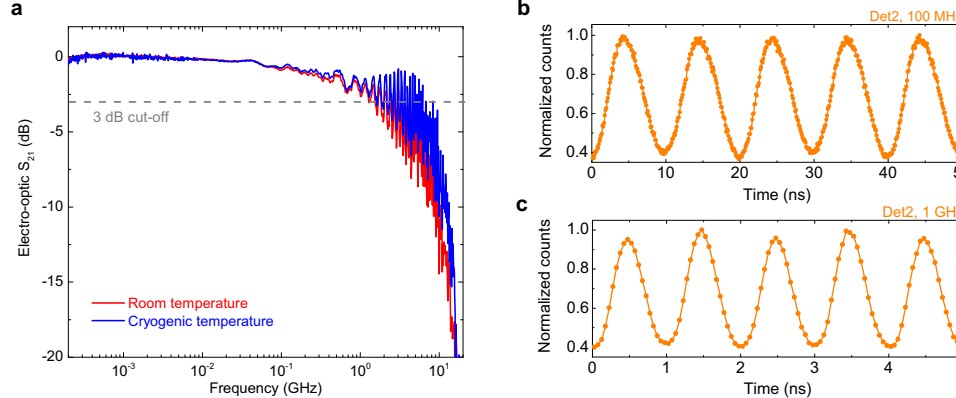

**Fig. 5 Combined operation of EOM and SNSPDs: high-speed operation. a** Modulation bandwidth of the EOM measured with a vector network analyzer at cryogenic temperature (blue trace) and room temperature (red trace) inside the same cryostat. The contribution of the RF lines of the cryostat is not calibrated out of the measurements. **b** Normalized count rates collected from Det2 with a time tagging module when the EOM is driven with a sinusoidal function with an amplitude of 5 V$_{pp}$ and a frequency of 100 MHz. Time bin width is set equal to 100 ps. **c** Normalized count rates collected from Det2 with a time tagging module when the EOM is driven with a sinusoidal function with an amplitude of 5 V$_{pp}$ and a frequency of 1 GHz. Time bin width is set equal to 50 ps.

**The combined operation of EOM and SNSPDs: high-speed operation**. To assess the high-speed performance of our device at cryogenic temperature, in Fig. 5a we report the bandwidth of the EOM measured at room- (red trace) and cryogenic temperature (blue trace) inside the same cryostat. Bandwidth measurements were conducted with a vector network analyzer (VNA) by coupling ≃1 mW optical power into the input of the MZI. Port 1 of the VNA was used to deliver a small-amplitude RF signal (200 kHz–18 GHz) to the EOM, while port 2 was connected to a fast photodiode (NewPort 1544-B) optically coupled to Out1. The modulation bandwidth plotted in the figure is the measured $S_{21}$ parameter properly normalized to its maximum value. The modulation bandwidth at cryogenic temperature was found to be slightly flatter than the one at room temperature, with an estimated 3 dB cut-off frequency ≃4 GHz for both cases, thus showing the possibility of performing high-speed operations in a cryogenic environment.

Single-photon detection at high modulation speed of the EOM was limited in our setup by electrical crosstalk between EOM and SNSPDs channels, introduced by the dual-purpose RF probe employed in our measurements, as well as by the close proximity of EOM and SNSPDs contact pads. The crosstalk was found to depend on the modulation frequency and started to become visible as a small oscillating noise at the readout of the detectors around a frequency of 1 MHz. Although the noise measured at the readout increases linearly with modulation frequency and reaches a magnitude larger than the SNSPD signal above 100 MHz—making any modulation around this frequency range seemingly impossible—, the actual noise current delivered to the detector is much lower. This is due to the kinetic inductance of superconducting nanowires, which naturally counteracts fast variations in their applied current[46]. Thus, upon removing the noise signal with the use of proper frequency filters, a "nanowire click" can be again visible at the readout of the detector and the normal functionality of the SNSPD is recovered.

Although electrical crosstalk still prevented us to operate the EOM at its full half-wave voltage, we were able to show single-photon detection at a modulation frequency up to 1 GHz with a reduced peak-to-peak amplitude of 5 V. A detailed analysis of electrical crosstalk in our setup, as well as an electric circuit model which precisely reproduces our experimental observations, can be found in Supplementary Note 2. Here, we report the main experimental results. High-speed modulation measurements were only performed using Det2, which was the SNSPD at a larger distance from the EOM and less affected by electrical crosstalk.

All the data described below were collected in a second measurement round, during which the detector was connected to low-noise room temperature amplifiers.

In Fig. 5b, c we report the time histograms for the photon counts collected from Det2, obtained for a modulation frequency applied to the EOM equal to 100 MHz, and equal to 1 GHz, respectively. Measurements were performed by biasing the detector at 85% of its critical current, with a setup similar to the one already described in the previous section. Attenuated laser light was coupled into In2 of the MZI, and the EOM driven with the output of a fast function generator amplified to a peak-to-peak amplitude of 5 V. The output trigger of the function generator was connected to the start channel of a time tagger, and the stop channel to the SNSPD output. For modulation at a frequency of 100 MHz, oscillating noise generated by electrical crosstalk was removed from the readout with a high-pass filter (140–2000 MHz) inserted between the amplification stage of the detector and the stop channel of the time tagger. For modulation at a frequency of 1 GHz, a low-pass filter (DC-500 MHz) was instead employed. The laser wavelength was set to the value for which the light at the output of the MZI is equally split between the two detectors for a zero applied voltage to the phase shifter. For both time histograms modulation visibility ≃60% could be measured, as correctly predicted given the applied voltage and the operating point of the EOM (see Fig. 4b).

## Discussion

We have reported the experimental demonstration of an ultra-low loss (0.2 dB/cm) reconfigurable LNOI waveguide network-integrated on-chip with single-photon detectors. The fabricated SNSPDs showed a nearly saturating detection efficiency at telecom wavelength, indicating a high internal quantum efficiency of our nanowires, and a dead time and timing jitter comparable with state-of-the-art superconducting nanowire detectors. Our EOM, characterized at cryogenic temperature, displayed a modulation bandwidth of ~4 GHz and a half-wave voltage in the range of 15 ÷ 20 V. For applications requiring only static tunability—e.g., a reconfigurable Boson sampler[49,50]—this voltage value can be readily achieved by the use of commercially available multi-channel digital-to-analog converters, while for fast switching operations at frequencies of hundreds of MHz up to the ~GHz regime, which is typically required for quantum photonic applications, a viable solution is the use of pulse generators combined with mid-power RF amplifiers[52].

A drawback of EOMs based on the Pockels effect is their relatively large footprint (1.7 mm in our implementation), which exceeds those of optomechanical systems[22]. However, we argue that the main obstacle for scaling up quantum photonic circuits is optical loss rather than the overall device dimensions. We estimate that the insertion loss of our EOM can be reduced to less than 0.1 dB with readily attainable improvements of our fabrication process (see the discussion in the Methods section), and similar values have already been reported for EOMs in LNOI waveguides[37]. Furthermore, due to the possibility of bending LNOI waveguides with a radius smaller than 100 μm, long optical circuits can be folded several times on the same chip. We anticipate that with our technology an integrated optical network made of more than 100 phase shifters can be fitted onto a sample with overall dimensions of $15 \times 15$ mm$^2$, providing a suitable platform for the implementation of noisy intermediate-scale quantum photonic processors[53,54].

Although electrical crosstalk prevented us to show high-speed modulation at the full half-wave voltage of the EOM, this problem might be overcome in future experiments with appropriate improvements of our setup (see the discussion in Supplementary Note 2). Alternatively, for fast switching operations in the ~GHz regime, the length of the modulator can be increased in order to achieve a lower $V_\pi$. Thus, after complementing fast optical switches and single-photon detectors with ultra-low loss integrated optical delay lines[55], our technology can also assist the development of universal quantum photonic processors by providing a powerful approach for the implementation of the spatial- and time- multiplexing schemes required for scalable linear optical quantum computing[26], as well as the manipulation of photonic cluster states via single-photon detection measurements with active fast feedforward[27].

## Methods

**Fabrication of the chip**. Our fabrication workflow is based on a top-down approach, which starts with the deposition of a ~5 nm thick NbTiN thin film (sheet resistance $\simeq 500 \, \Omega$ sq) on a 300 nm thick X-cut LiNbO$_3$ film bonded on 4.75 μm thick SiO$_2$ layer thermally grown on a silicon substrate (wafer produced by NanoLN). The deposition is performed by DC-magnetron sputtering using a Nb/Ti alloy target in an Ar/N$_2$ atmosphere. This sputtering process is carried out at room temperature in an ultra-high vacuum environment, making the deposition of superconducting films compatible also with temperature-sensitive substrates[45].

To realize a photonic chip where modulators and waveguide-integrated single-photon detectors are monolithically integrated into a single device, several lithography exposures and etching processes are required. As a very first step, the contact pads needed for the read-out of the detectors signals and the alignment markers (5 nm Cr/80 Au/5 nm Cr, deposited via physical vapor deposition) are formed by using PMMA as a positive tone resists for a lift-off process. Secondly, to fabricate the detectors, we make use of a 100 keV electron beam lithography (EBL) system to expose HSQ resist, which acts as a mask during the etching of the NbTiN film in a SF$_6$ + Ar plasma. To further process the sample and to protect the nanowires from oxidation during the following EBL and dry etching steps, we cover the SNSPDs with a 200 nm thick and 750 nm wide SiO$_2$ protection layer (see Fig. 1d), which is deposited by RF-sputtering and patterned by EBL and dry etching in CHF$_3$ + Ar chemistry. We added Argon to the fluorine gases during the etching of both the SNSPDs and the SiO$_2$ cover layer for minimizing the formation of non-volatile LiF by-products on the plain LN surface. Subsequently, we fabricate the optical circuitry. First, we expose a negative-tone resist (ArN7520) in EBL, which allows us to transfer the pattern into the LN thin film using an Oxford100 ICP-RIE tool, where we fully etch the waveguides in a highly energetic and low-pressure Ar plasma. Unlike other photonic platforms that can be etched with purely chemical dry processes, so far only physical sputtering in inert gas was proven to be successful in etching LiNbO$_3$ waveguides featuring smooth surfaces[36]. As a side effect, the waveguides show a typical sidewall angle of approximately 62° (measured with an AFM) and etched material is redeposited on the waveguide sidewalls during the physical sputtering process. Therefore, to achieve low propagation loss, after waveguide etching and resist stripping, we also remove the sidewall redeposition. The use of an RCA-1 cleaning solution has been proven effective to remove the sidewall redeposition[56], but we have observed that it can also lift the SiO$_2$ cover and etch the detectors. Therefore, an additional protection layer able to withstand this wet process was employed. Successful was the use of a thick electrically cured HSQ cover, which is approximately 35 μm wider than our waveguide-integrated SNSPDs from all sides (see Fig. 1a), so that the redeposition

can be completely removed over the entire length of the waveguides, except for the area in close proximity of the wire.

After waveguide fabrication, the device is clad with a 750 nm thick HSQ layer, which is cured everywhere by an electron beam exposure except in the area occupied by the contact pads of the detectors (see Fig. 1a), where it is washed away by development in an MF-319 solution. As a very last step, we carry out the fabrication of the modulators by following the same procedure as that employed for the SNSPD contact pads and alignment markers.

**Propagation loss**. To determine the loss of our LN waveguides both at room- and cryogenic-temperature, we measured the transmission spectra of several racetracks- and ring- resonators, where the gap between the cavity and the bus waveguide was varied. The straight arm of the racetrack resonators is 500 μm long, and in both structures the bend radius is set to $R = 70$ μm. We average the measured linewidths of several resonances in the wavelength range of 1530–1550 nm, and, by comparing the extracted quality factors of the ring- and racetrack-resonators in the critical coupling regime, we estimate the propagation and bending loss.

To determine the absorption loss induced by the signal and ground electrodes of our EOM, we also measured the quality factor of racetrack resonators where two 500 μm long gold stripes were patterned on top of the HSQ cladding on both sides of one of the two straight arms. At room temperature, we extrapolated a metal-induced absorption loss of ~5 dB/cm. This value fits well with the insertion loss measurement reported in Fig. 2c and indicates a negligible contribution to the total insertion loss coming from propagation in our waveguides and crossing with the metal electrodes. A lower metal-induced absorption loss equal to ~3.7 dB/cm was measured at cryogenic temperature.

A first improvement of the insertion loss of our EOM can be readily obtained by increasing the gap between signal and ground electrodes (equal to 1.7 μm in our implementation). This way, the metal-induced absorption loss can be reduced exponentially at the expense of a linear increase in the half-wave voltage of the device. We further note that for our EOM configuration high absorption loss is induced by the high refractive index of the chromium adhesion layer used for the fabrication of gold electrodes. We numerically estimate that by increasing the gap between the electrodes to 2 μm and by using an adhesion layer with a lower refractive index—e.g., aluminum—, the insertion loss of our EOM can be reduced to less than 0.1 dB, while maintaining a half-wave voltage $V_\pi < 20$ V.

As higher-order modes of a waveguide are typically excited only in the proximity of a bend, an additional improvement of our electrodes configuration might be achieved by adiabatically increasing the waveguide width—here designed equally to 1.1 μm to ensure single-mode operation—in the straight sections. We expect that this would further reduce the effect of metal-induced absorption loss, enable to achieve a lower half-wave voltage, and enlarge the modulation bandwidth thanks to the wider gap between the electrodes.

**On-chip detection efficiency**. With reference to Fig. 1a, we denote $P_{in}$ as the optical power at the input of the grating coupler of In2, and $P_{out}$ the power measured from either Out1 or Out2 when the laser wavelength is set for constructive interference such that all the light entering the MZI is directed toward one of the two outputs. By assuming equal coupling efficiency for the input and output grating couplers, the photon flux at the input of either Det1 or Det2 can be calculated as

$$\Phi = \frac{\lambda}{hc} \sqrt{\frac{P_{in}P_{out}L}{(1-S)}} S,$$

where $\lambda$ is the wavelength of the incoming photons, $c$ the light speed in a vacuum, $h$ the Planck's constant, $L$ is the insertion loss of the MZI, and $S$ is the splitting ratio of the directional coupler immediately prior to the detector. This way, the input laser power $P_{in}$ can be attenuated to get a calibrated photon flux $\Phi = 10^6$ photons/s at the input of the SNSPDs and the on-chip detection efficiency measured as $OCDE = (CR - DCR)/\Phi$, where CR is the count rate measured by the detector and DCR the dark count rate.

## Data availability

The source data underlying Figs. 2–5 are provided as a Source Data file. All other data supporting this study are available from the corresponding authors upon reasonable request. Source data are provided with this paper.

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

## Acknowledgements

F.L. acknowledges the support of the Humboldt Research Fellowship for Postdoctoral Researchers. We would like to thank the Münster Nanofabrication Facility (MNF) for their support in nanofabrication-related matters. We acknowledge support from the European Union's Horizon 2020 Research and Innovation Action under grant agreement no. 899824 (FET-OPEN, SURQUID). C.S. acknowledges support from the Ministry for Culture and Science of North Rhine-Westphalia (421-8.03.03.02–130428). This project has received funding from the European Research Council (ERC) under the European Union's Horizon 2020 research and innovation program (grant agreement No. 724707).

## Author contributions

E.L. and F.L. developed the fabrication workflow and fabricated the integrated device. M.A.W. developed the process for the deposition of superconducting films on an LN substrate. E.L., F.B., M.A.W., and F.L. performed the experimental measurements. S.F. contributed to the development and testing of EOMs in LNOI waveguides, and to the design of integrated SNSPDs. F.L. designed the integrated device and coordinated the project. C.S. and W.H.P.P. supervised the project. E.L. and F.L. wrote the paper with contributions from all authors.

## Funding

## Competing interests

The authors declare no competing interests.
