## [Peer Review File · Nature Communications]

Reviewers' Comments:

Reviewer #1:

Remarks to the Author:

In the article by Emma Lomonte et al. entitled "Single-photon detection and cryogenic reconfigurability in Lithium Niobate nanophotonic circuits", the authors demonstrate the combined operation of an electrically tunable Mach-Zehnder interferometer together with two superconducting nanowire single photon detectors (SNSPDs) integrated on a Lithium-Niobate-On-Insulator (LNOI) optical circuit. This platform is emerging due to several properties: high second-order nonlinearity, small waveguide footprint, and low propagation loss in a broad wavelength range. The capability to operate those reconfigurable circuits even at low temperature, allowing in this way their use with integrated SNSPDs, enables the realization of photonic devices for active manipulation and detection of quantum states of light. Moreover, in the article the authors demonstrate an optical modulation 3 dB cut-off frequency in the range of ≈ 4 GHz, and a decay time of the SNSPD (related to the maximum operating frequency) of about 6 ns, comparable to state-of-the-art detectors. The Mach-Zehnder interferometer has been chosen as it is the fundamental building block for the implementation of reconfigurable optical networks.

This is the first demonstration of the integration of single photon detectors and optical manipulation on chip on a Lithium-Niobate-On-Insulator platform. Several groups are working on the implementation of reconfigurable optical networks to achieve manipulation and detection of quantum states of light on different platforms: for this reason the results here obtained will be of interest to a wide community.

The results here presented are original and they will be a reference point for the community. The references are appropriate: the paper correctly reports all the previous works on this argument.

The reported data are complete and well-presented throughout the article. The experimental achievements are robust, the data are valid and the article is reliable in all its parts: the abstract, the introduction and the final discussion are clear and well written.

For all those reasons I recommend the article for publication.

However, I suggested some improvements and minor revisions that could help the authors to increase further the article quality.

- Row 133

The propagation loss of our waveguides was characterized both at room- and cryogenic-temperature by measuring the Q-factor of ring- and racetrack- resonators fabricated on the same chip...

It could be useful for the reader to have an image of the optical circuit (ring and/or racetrack) as an inset in fig 2.

- Row 136

The losses are well above the reported results from previous work. For example in ref. 35 "Monolithic ultra-high-Q lithium niobate microring resonator" the losses are 2.7 dB/m. Could the author infer the source of this additional losses with respect to previous results?

- Row 147 and fig. 2

Fig.2a shows a critically coupled ring resonator meanwhile Fig.2b shows a ring that is either overcoupled or undercoupled (in methods, row 329 the author states that: "...by comparing the extracted quality factors of ring- and racetrack-resonators in the critical coupling regime, we estimate the propagation and bending loss"). To this two conditions (overcoupled or undercoupled) correspond different losses. How can the authors be sure of the condition of the ring and infer the values for QL and Q in row 147?

- Row 166

"We attribute this imperfect detection efficiency to either an incomplete etching of the

superconducting film in the detector region or to the presence of redeposition...". An incomplete etching would probably give a non saturated behaviour of the detector due to the internal efficiency lower than 1. Could the authors comment further on this point?

- Row 171

Please put a vertical line in the graph of fig. 3a) and b). You refer to this value also in line 199 and line 212.

- Row 222

"...indicating only a small reduction of the electro-optic coefficient of LNOI waveguides at 222 cryogenic temperatures." Is the change of V_{π} due to the EO coefficient or to the thermo-optic coefficient which could change the field distribution in the waveguide? Could the authors give their comment on this discrepancy?

- Row 238

Which fast photodiodes have you used? How have you biased the EOM at high modulation speeds?

Reviewer #2:

Remarks to the Author:

The manuscript by Lomonte et al. reports the heterogeneous integration of SNSPDs on LNOI electro-optic modulators. The combination of these two technologically important components is of particular interest to the integrated quantum photonics community. The authors successfully integrated them on the same chip with specially tailored fabrication processes. They performed detailed characterization of individual components and showed preliminary combined operation of the two. The manuscript is well written with a good introduction and detailed description of the fabrication process and measurement results.

However, despite the strategic importance of the work, I find a few critical issues. First, the performance of the LNOI modulator is significantly worse than state of the art. In fact, a 15-20 V V_{π} and few-GHz bandwidth almost demotivate one's interest in using LNOI modulators for integrated quantum photonics. Second, the LNOI waveguide integrated SNSPDs do not outperform what has been demonstrated in the literature either (OCDE of 25% vs 46%) [Appl. Phys. Lett. 116, 151102 (2020)]. Most importantly, the combined operation does not show any desirable advantages. It only worked up to ~ 100 kHz (data not shown). This is largely due to the electrical talk, and partly due to the large V_{π} required for modulator operation. The authors describe this issue as "merely constitutes a technical difficulty." However, I think this is a critical issue that deserves much attention and may not be as easily addressable as what the authors predicted.

In my opinion, the impact, novelty, device performance, and quality of the work do not meet Nature Communications' standard. I, therefore, do not recommend its publication in Nature Communications. Below are some detailed technical comments.

1. The electrode design of the LN modulator seems to be inefficient. The small gap will give large capacitance, and V_{π} is worse than the standard case where metal sits on the partially etched LN waveguide slab. It is better to compare the current design to standard approaches and propose some future improvements.

2. When comparing Q-factors at cryo and room temperature, Figures 2 (a) and (b) showed resonances measured at very different wavelengths. Why not make a direct comparison at the same resonance? Also, in Figure 2(c), is the 0.82 dB insertion loss consistent for all wavelengths (i.e. other FSRs peaks)?

3. When describing modulator frequency response (Line 242), the authors attribute the large EO S_{21} oscillation to impedance mismatch of the RF lines in the cryostat. But shouldn't a proper VNA calibration factor out the impedance mismatch and give the de-embedded response up to the probe? Also, "large bandwidth oscillation" should be "large EO S_{21} oscillation." And Figure 4d y axis should be "Electro-optic S_{21} (dB)" to be clear, I assume.

4. When describing SNSPDs in Line 156, could the authors be more explicit about what the “novel fabrication approach” is? And the authors attribute imperfect detection efficiency to incomplete NbTiN etching or redeposition on the sidewall. Wouldn't incomplete etching of NbTiN cause large propagation loss of the entire waveguide (which will be observed as large insertion loss), and how redeposition would affect NbTiN absorption?

5. Figure 4 and Line 230: the authors attribute the lower extinction ratio to imperfect splitting ratio of the directional couplers. Which directional coupler is referred to? Why Det2 showed better extinction? Also, with DC bias, do the authors observe any bias drift/DC stability issue of the modulator?

6. Line 279: the authors propose that wire bonding SNSPD and modulator to two PCBs will solve the problem of crosstalk. I am skeptical about this. Is there any preliminary evidence? The SNSPD is sensitive to orders of magnitude smaller voltage than the modulator drive, they are in close proximity, and they will likely share the same ground. It doesn't seem to be an easy problem to solve. Some preliminary results will be more convincing.

Reviewer #3:

Remarks to the Author:

This is a very interesting and potentially important work: implementing high-efficiency and high-speed optical switches and single-photon detectors on the same photonic circuit. If successful, this can have important implications for quantum photonics. The authors have done a very good job developing the device and the work is precisely communicated in the manuscript.

My main concern is that the work appears somewhat unfinished therefore not yet exploiting the competitive advantage that the approach may have compared to other methods (nano-mechanical, for instance).

While SNSPDs and EOM switches have been implemented before on the LNOI platform, the main novelty is that they are combined. Ideally this would allow very rapid switching on quantum pulses. Unfortunately the authors do only demonstrate explicit modulation at 1 kHz, and high-speed operation is limited by cross-talk in the device. It is argued that this cross-talk can be straightforwardly reduced, however, an explicit demonstration would be a major asset. Furthermore, the SNSPD detector efficiencies are rather low and coupling losses to the device are high. This leaves the impression that there is still a lot of work to be done before the promised quantum-photonics applications can be realized.

I would encourage the authors to complete their work and show high-speed switching and efficient single-photon detection on the platform. I think this would be an important result for the community warranting publication in Nat. Comm.

We would like to thank all the Referees for their careful assessment of our manuscript and the very helpful suggestions on how to improve the presentation. We have followed these recommendations in full and have revised the text and the figures accordingly. We have also carried out new experiments and added substantial new data to both the main text and the supplementary materials. We hope that these changes and clarifications will answer all questions of the referees. In the following, we provide a detailed response to each comment.

Response to Reviewer 1:

Comment 1 (row 133). *“The propagation loss of our waveguides was characterized both at room- and cryogenic- temperature by measuring the Q-factor of ring- and racetrack- resonators fabricated on the same chip.” It could be useful for the reader to have an image of the optical circuit (ring and/or racetrack) as an inset in fig 2.*

Answer 1. We thank the referee for this suggestion. As an inset of Fig.2 a,b, we added a sketch of the racetrack resonator device used to measure the resonances reported in panel a,b at room- and cryogenic-temperature, respectively. The caption of the figure has been also changed accordingly.

Comment 2 (row 136). *The losses are well above the reported results from previous work. For example in ref. 35 “Monolithic ultra-high-Q lithium niobate microring resonator” the losses are 2.7 dB/m. Could the author infer the source of this additional losses with respect to previous results?*

Answer 2. We would like to clarify that in Ref.35 “Monolithic ultra-high-Q lithium niobate microring resonator”, propagation loss as low as 2.7 dB/m was measured for multimode waveguides with a width of 2.4 μm . In the same paper, for a waveguide width of 1 μm , the propagation loss increases to around 0.1 dB/cm. This is a value comparable to the one measured for our waveguides (0.2 dB/cm), where a width of 1.1 μm was chosen to ensure single-mode operation in the 1550 nm wavelength range. We clarified this point in line 142-143 of the new manuscript:

*“We note that the extracted value of propagation loss is close to the *status quo* for single-mode LNOI waveguides (waveguide width $\simeq 1 \mu\text{m}$)³⁶.”*

Comment 3 (row 147 and Fig.2). *Fig.2a shows a critically coupled ring resonator meanwhile Fig.2b shows a ring that is either overcoupled or undercoupled (in methods, row 329 the author states that: “...by comparing the extracted quality factors of ring- and racetrack-resonators in the critical coupling regime, we estimate the propagation and bending loss”). To this two conditions (overcoupled or undercoupled) correspond different losses. How can the authors be sure of the condition of the ring and infer the values for QL and Q in row 147?*

Answer 3. The Reviewer is indeed correct. Although critical coupling was achieved at room temperature, the same does not hold at cryogenic temperature. Since the racetrack resonator is either undercoupled or overcoupled, this effectively introduces an error in the inference of the propagation loss. We clarified this point in line 140-142 of the revised manuscript:

“While at room temperature critical coupling was achieved, at cryogenic temperature a slightly reduced extinction ratio of the measured resonances introduced an estimated error of around ± 0.05 dB/cm in the inference of the propagation loss.”

Comment 4 (row 166). *“We attribute this imperfect detection efficiency to either an incomplete etching of the superconducting film in the detector region or to the presence of redeposition...”. An incomplete etching would probably give a non saturated behaviour of the detector due to the internal efficiency lower than 1. Could the authors comment further on this point?*

Answer 4. Etching residuals are typically present as small clusters rather than in the form of a continuous film. This is why we believe that NbTiN residuals might be present without affecting the performance of the nanowires. We now report in Section I of Supplementary Information a detailed analysis of the source of inefficiency of our detectors, to which we make explicit reference in line 176-178 of the new manuscript.

Comment 5 (row 171). *Please put a vertical line in the graph of fig. 3a) and b). You refer to this value also in line 199 and line 212.*

Answer 5. The plots have been modified according to the Reviewer's suggestion. A new version of the figure can be found in the revised manuscript. The caption of Fig.3 has also been changed accordingly:

“Fig. 3. Performance of the waveguide-integrated detectors. a-b Measured on-chip detection efficiency (OCDE) of the two detectors as a function of the bias current. **The dashed-dotted line indicates the value of the current $I=0.85 \cdot I_c$, at which the two detectors were biased to measure the timing jitter and to show the joint operation of the EOM and the SNSDPs.”**

Comment 6 (row 222). *“...indicating only a small reduction of the electro-optic coefficient of LNOI waveguides at cryogenic temperatures.” Is the change of V_{π} due to the EO coefficient or to the thermo-optic coefficient which could change the field distribution in the waveguide? Could the authors give their comment on this discrepancy?*

Answer 6. We thank the referee for raising this point and attribute the change in V_{π} to a reduced r_{33} coefficient, as also found in other works. We address the comment of the Reviewer in line 232-235 of the new manuscript:

“We attribute this change in the half-wave voltage of the EOM to a slight reduction of the r_{33} coefficient of LN at cryogenic temperature, an effect already observed in bulk LN crystals³¹ as well as LN waveguides fabricated by Ti-indiffusion³².”

Comment 7 (row 238). *Which fast photodiodes have you used? How have you biased the EOM at high modulation speeds?*

Answer 7. We carried out the measurements with a commercial high-speed photodetector (see below). We measured the bandwidth of the EOM by using a vector network analyzer (VNA). Port1 of the VNA was used to drive the EOM with a small-amplitude RF signal (200 kHz-18 GHz), while Port2 of the VNA was connected to a fast photodiode (NewPort 1554-B) optically coupled to Out1. In line 280-289 of the new manuscript, we now provide a more detailed description of our bandwidth measurements.

We hope to have answered all questions of the referee satisfactorily. We hope that with the additional experimental data and the revisions to the manuscript she/he will support publication of our results.

Response to Reviewer 2:

Comment 1. *Most importantly, the combined operation does not show any desirable advantages. It only worked up to ~100 kHz (data not shown). This is largely due to the electrical crosstalk, and partly due to the large V_{pi} required for modulator operation. The authors describe this issue as “merely constitutes a technical difficulty.” However, I think this is a critical issue that deserves much attention and may not be as easily addressable as what the authors predicted.*

Answer 1. We appreciate the comment of the referee and have now included new experimental data showing high-speed operation of our devices.

In this revised version of our manuscript, we have conducted a deeper analysis of electrical crosstalk in our setup and found out that it is less of a concern than what we initially evaluated. Although the noise measured at the readout of the detectors increases linearly with modulation frequency, and reaches a magnitude larger than the SNSPD signal above 100 MHz – making any modulation in this frequency range seemingly impossible – the actual noise current delivered to the SNSPD is much lower. This is due to the kinetic inductance of superconducting nanowires, which naturally counteracts fast variations in their applied current. Thus, upon removing the noise signal with the use of proper frequency filters, a “nanowire click” can be again visible at the readout of the detector and the normal functionality of the SNSPD is recovered.

Although electrical crosstalk still makes operation of the EOM at his full half-wave voltage challenging, we now explicitly show single-photon detection at a modulation frequency up to 1 GHz with a reduced peak-to-peak amplitude of 5 V (around one third of the half-wave voltage). Our results on high-speed modulation measurements are reported in Fig. 5b,c of the new manuscript, and commented in lines 300-331.

We have also complemented our paper with a Supplementary Information material where we analyze in detail the problem of electrical crosstalk, present an electric circuit model which precisely reproduces our experimental observations, and propose viable ways for overcoming this issue in future experiments by backing up our claims with systematic sets of data.

Comment 2.

“The electrode design of the LN modulator seems to be inefficient. The small gap will give large capacitance, and $V_{pi} \cdot L$ is worse than the standard case where metal sits on the partially etched LN waveguide slab. It is better to compare the current design to standard approaches and propose some future improvements.”

And:

“First, the performance of the LNOI modulator is significantly worse than state of the art. In fact, a 15-20 V V_{pi} and few-GHz bandwidth almost demotivate one’s interest in using LNOI modulators for integrated quantum photonics.”

Answer 2. We believe that certain points of our paper might have been not correctly interpreted and thus the Reviewer’s concerns are misplaced. Specifically, we would like to point out:

- At room temperature, our EOM displays a $V_{pi}=15.5$ V with 1.7 mm long electrodes, which corresponds to a $V_{pi} \cdot L=2.6$ V*cm. This voltage-length product compares well with configurations where signal and ground electrodes sit at the two sides of the waveguide. For example, the EOM demonstrated in Ref. 37 (C. Wang et al., Nature 562, 101-104 (2018)) displayed a $V_{pi}=1.4$ V using 20 mm long electrodes ($V_{pi} \cdot L=2.8$ V*cm). In general, voltage length products of the order of ≈ 2.5

$V \cdot \text{cm}$ appear to be quite a standard in all the recently published works (see, e.g., Ref. 48 of the new manuscript). The only EOM that more pronouncedly outperforms our device is the one of Ref. 42, which reported a $V_{\pi} \cdot L = 1.8 \text{ V} \cdot \text{cm}$ (however, the 2 mm long EOM reported in that work also displayed a large insertion loss of around 2 dB).

We have clarified this point in line 230-235 of the new manuscript:

“By fitting the normalized count rates of the two detectors with a sinusoidal function we extracted a half-wave voltage at cryogenic temperature $V_{\pi}^{\text{cryo}} = 17.8 \text{ V}$ (see Fig. 4b). By comparison, the half-wave voltage measured at room temperature using two standard photodiodes and the same ramp function was found equal to $V_{\pi}^{\text{RT}} = 15.5 \text{ V}$. We attribute this change in the half-wave voltage of the EOM to a slight reduction of the r_{33} coefficient of LN at cryogenic temperature, an effect already observed in bulk LN crystals³¹ as well as LN waveguides fabricated by Ti-indiffusion³². Given the length of the electrodes of our EOM (1.7 mm), we estimate voltage-length products $V_{\pi} \cdot L \simeq 2.6 \text{ V} \cdot \text{cm}$ at room temperature, and $V_{\pi} \cdot L \simeq 3 \text{ V} \cdot \text{cm}$ at cryogenic temperature.”

There are in addition several improvements that could be still done to our electrodes configuration. A discussion can be found in lines 412-423 of the Methods section.

We have already stated in the previous version of the manuscript (see lines 97-100) why we opted for this electrode configuration: this is to enable a lossless crossing of the electrodes with the waveguides, without the need of additional fabrication steps which would further increase the complexity of an already highly challenging fabrication process.

- In all the envisaged quantum photonic applications which are listed in our paper, the EOM must be only operated at frequencies ranging from few hundreds of MHz up to around 1 GHz, and it would be pointless optimizing our device for modulation bandwidths much larger than this value. Thus, the small gap between the electrodes does not represent a major concern. We have clarified this point in line 50-54 of the new manuscript:

“Moreover, opto-mechanical devices suffer from a maximum modulation bandwidth limited to the \sim MHz range, which prevents their usage for a large number of important applications – such as quantum computing protocols based on time-bin encoding^{23,24}, spatial- and time-multiplexing schemes for scalable quantum computing^{2,25,26}, or fast feedforward operations for measurement-based quantum computation^{27–29} – where a bandwidth in the range of few hundreds of MHz up to the \sim GHz regime is mandatory.”

To add some specific examples: in quantum computing protocols based on time-bin encoding the maximum achievable modulation rate is set by the dead time of the single-photon detectors. For standard waveguide-integrated SNSPDs this is typically of the order of few nanoseconds, and it can be pushed down to around 500 ps by the use of cavity structures (see, e.g., Ref.12,13). For the case of spatial multiplexing schemes and fast feedforward operations for measurement-based quantum computing, the state of the EOM must be quickly tuned in response to a single-photon detection event. The time delay between single-photon detection and electro-optic modulation is usually limited by electronic processing time rather than by the bandwidth of the modulator, and it can be probably pushed down to few nanoseconds using fast FPGAs (see, e.g., E. Meyer-Scott et al., Rev Sci Instrum 91(4), 041101 (2020) for a discussion).

- In our experiment we targeted on purpose a V_{π} in the range of 15-20 V (indeed, the measured half-wave voltage is consistent with our numerical prediction) as a trade-off between driving voltage and device footprint, in order to enhance the spatial scalability of our system. Importantly, the reduced dimension of our electrodes makes our platform also suitable for several applications in quantum photonics where only static over fast reconfigurability is required, and large optical networks made of several phase shifters (a so called Reck scheme) are needed to implement

arbitrary linear optical operations on quantum states of light. In this revised version of our manuscript we have tried to put more emphasis on this class of applications, also in view of the bias-drift-free operation of our EOM (see Answer 5).

For applications requiring fast switching operations at frequencies ranging from few hundreds of MHz up to around 1 GHz, this voltage value can be achieved by the use of pulse generators combined with mid-power RF amplifiers (we already made this point in the previous version of the manuscript, see line 338-342 in Discussion). Such systems were for instance developed in Ref. 3,52 for driving multi-channels EOMs implemented in proton-exchanged waveguides, with specific target to quantum photonic applications.

The length of the modulator can be anyway still increased – at the expense of a reduced spatial scalability - in order to achieve a lower half-wave voltage.

Comment 2. *When comparing Q-factors at cryo and room temperature, Figures 2 (a) and (b) showed resonances measured at very different wavelengths. Why not make a direct comparison at the same resonance? Also, in Figure 2(c), is the 0.82 dB insertion loss consistent for all wavelengths (i.e. other FSRs peaks)?*

Answer 2. We followed the Reviewer's suggestion and are now showing two resonances taken at a closer wavelength. Unfortunately we are not able to show two resonances taken at exactly the same wavelength, as several of them display a double-peak shape. This is a well-known effect often observed in waveguides characterized by low propagation loss, due to the excitation of clockwise and anticlockwise propagating modes in the resonator. The loaded Q-factor is only evaluated in resonances with a single peak, where a fit with a Lorentzian function is more accurate. Anyway, given the narrow wavelength range in which the resonances are taken, we believe that it is safe to assume that a similar value of propagation loss is present both at room- and cryogenic temperature.

Regarding the insertion loss of the MZI: it is not possible to make a precise estimation of the insertion loss at all wavelengths, as different gratings typically display a slightly different central coupling wavelength (this can also be visually appreciated from the plot of Fig. 2c). The insertion loss was evaluated by fitting the spectrum of the Reference DC with a Gaussian function, the spectrum of the outputs of the MZI with a Gaussian function convoluted with a sinusoidal function, and by comparing the maxima of the two Gaussians. To take into account the error caused by variations in the coupling efficiency of different gratings, the insertion loss is averaged over several identical devices. We clarified in this point in the caption of Fig. 2c:

“The insertion loss of the MZI is estimated by fitting the transmission spectrum of the reference DC with a Gaussian function, the spectrum of Out1/Out2 with a Gaussian function multiplied by a sinusoidal function, and by comparing the maxima of the two Gaussians. The insertion loss averaged over six identical devices fabricated on the same chip is found equal to (0.82 ± 0.24) dB.”

Comment 3. *When describing modulator frequency response (Line 242), the authors attribute the large EO S21 oscillation to impedance mismatch of the RF lines in the cryostat. But shouldn't a proper VNA calibration factor out the impedance mismatch and gives the de-embedded response up to the probe? Also, “large bandwidth oscillation” should be “large EO S21 oscillation.” And Figure 4d y axis should be “Electro-optic S21 (dB)” to be clear, I assume.*

Answer 3. The text and the label of the y axis of Fig. 5a (no longer 4d in this new version of the manuscript) have been changed according to the Reviewer's suggestion.

Regarding the bandwidth oscillations at high frequencies: we were unable to cancel them out even after subtracting the contribution of the RF lines of the cryostat. Although we can still confirm that

they are not present when the measurement is repeated in a different setup, since we are currently unable to precisely identify the source of the problem, we have deleted from the text our claim that they were caused by an impedance mismatch of the RF lines. Anyway, we think that it should be correct to show the measured bandwidth without subtracting the contribution of the RF lines, since this one is the condition in which our device is effectively operated inside the cryostat. We now specify in the caption of Fig. 5a that the contribution of the RF lines is not calibrated out of the bandwidth measurement.

Comment 4. *When describing SNSPDs in Line 156, could the authors be more explicit about what the “novel fabrication approach” is? And the authors attribute imperfect detection efficiency to incomplete NbTiN etching or redeposition on the sidewall. Wouldn't incomplete etch of NbTiN cause large propagation loss of the entire waveguide (which will be observed as large insertion loss), and how redeposition would affect NbTiN absorption?*

Answer 4. The main challenge for the realization of waveguide-integrated SNSPDs on a LNOI platform is the necessity of employing wet etching processes (in our case, RCA-1 clean) for removing redeposition of sputtered material on the waveguide sidewalls during Ar etching of the LN film. This makes the fabrication of LNOI waveguides not compatible with conventional methods for fabrication of superconducting nanowires, which are usually based on a top-down approach (i.e., the nanowire is patterned as a very first step before waveguide fabrication).

To overcome this problem, in Ref. 40 Sayem et al. employed a bottom-up approach (i.e., the nanowire is patterned on top of the waveguide as a last fabrication step). However, they also required the deposition of a HfO_2 thin buffer layer on top of the waveguides for protecting them from the adverse effects of fluorine etching during the patterning of the nanowires. The use of this buffer layer was responsible for the elevated propagation loss (5 dB/cm) of the fabricated photonic circuits.

In our case, we employed a more conventional top-down approach and used an electrically cured HSQ cover (approximately 35 μm wider than the nanowire from all sides) to protect the nanowires from wet etching during RCA-1 clean (see the Fabrication section in Methods for more details).

We cannot say much more about the exact fabrication challenges encountered in Ref. 40 and if they could have been overcome in a different way. However we now clarify in the text (lines 160-168) which is the main difference of our fabrication approach:

“Only recently, the first SNSPDs integrated with LNOI waveguides were reported, using a ~ 5 nm niobium nitride (NbN) thin film as a superconducting material⁴⁰. Although an on-chip detection efficiency (OCDE) $\simeq 46\%$ was achieved, the waveguides showed elevated levels of propagation loss (5 dB/cm) due to the employed process for fabrication of superconducting nanowires **based on a bottom-up approach**. **In this work** we use a **top-down** fabrication for LNOI waveguide-integrated SNSPDs and achieve substantially lower propagation loss (0.2 dB/cm). For our experimental implementation, the use of niobium titanium nitride (NbTiN) nanowires was chosen over NbN because of their low kinetic inductance^{43,44}, which enables high speed detection, and low dark count rate¹¹. Its polycrystalline phase also eases the growth on several substrates, granting high uniformity and high yield production even for very thin films⁴⁵.”

We also would like to stress again that, although SNSPDs with a 46% detection efficiency were previously demonstrated in Ref. 40, the waveguides showed elevated levels of propagation loss (5 dB/cm) because of the employed process for fabrication of superconducting nanowires. Our detectors display a reduced efficiency of around 25%, yet are integrated with waveguides with considerably lower propagation loss (0.2 dB/cm).

Regarding the source of inefficiency of our detectors: we believe that both potential residuals of the NbTiN films, as well as redeposition of sputtered material on the waveguide sidewalls (which is not removed in the detector region because of the employed HSQ cover) might cause elevated propagation loss of the waveguide in the proximity of the nanowires. Such large propagation loss cannot be present in the remaining optical circuit, where both sidewall redeposition and potential residuals of the NbTiN films are removed during RCA-1 clean.

In Section I of the Supplementary Information, we now report a detailed discussion on the source of inefficiency of our nanowires and present new experimental data on the measurement of the nanowires absorption. We make explicit reference to this section in line 174-177:

“We attribute this imperfect detection efficiency to either an incomplete etching of the superconducting film in the detector region, or to the **redeposition of sputtered material on the waveguide sidewalls during Ar etching of the LN film which is not removed in proximity of the nanowires (see the Fabrication section in Methods, and Section I of the Supplementary Information for more details).**”

Comment 5. *Figure 4 and Line 230: the authors attribute the lower extinction ratio to imperfect splitting ratio of the directional couplers. Which directional coupler is referred to? Why Det2 showed better extinction? Also, with DC bias, do the authors observe any bias drift/DC stability issue of the modulator?*

Answer 5. We thank the referee for bringing this up. We refer to the splitting ratio of the two directional couplers of the MZI. We clarified this point in line 242-244:

“The lower extinction ratio (≈ 15 dB) observed for Det1 is due to an imperfect 50:50 splitting ratio of the **two directional couplers of the MZI**, which we determined equal to $\approx 56\%$ from our measurements.”

When laser light is injected into In2, and a pi phase shift is induced between the two arms of the MZI, all the output light is completely directed to Det1 independently from the splitting ratio of the two DCs. Thus, the extinction ratio measured on Det2 is only limited by the signal-to-noise ratio of the detectors. Conversely, when no phase shift is induced between the two arms, all the output light is completely directed to Det2 only in the ideal case of two directional couplers with a 50:50 splitting ratio. Thus, the extinction ratio measured on Det1 is lower. This is a result that can be easily verified with a transfer matrix formalism. The 56% splitting ratio is measured from the Reference DC, which has the same coupling length of the two DCs of the MZI.

Regarding the DC-bias-drift of our EOM:

This is a very interesting result that we did not initially include in our paper and that will deserve a deeper investigation in the future. Indeed, the DC-bias-drift is a well-known issue of EOMs implemented in LN substrates, and this problem has been recently reported also for LNOI waveguides (see Ref. 48 of the new manuscript). At room temperature also our device suffers from large instabilities, at the point that it can be effectively operated only with an AC voltage. Remarkably, the problem seems to completely disappear as the sample is cooled down to cryogenic temperature. In Fig. 4d of the new manuscript we now report a stability test performed by recording the photon counts from Det2 over a time of 12 hours at an applied DC bias of 16.5 V. The recorded signal is found highly stable in time, with an overall variation of less than 0.05 dB over the measurement time window. We comment this result in line 258-278 of the new manuscript.

Comment 6. *Line 279: the authors propose that wire bonding SNSPD and modulator to two PCBs will solve the problem of crosstalk. I am skeptical about this. Is there any preliminary evidence? The SNSPD is sensitive to orders of magnitude smaller voltage than the modulator drive, they are*

in close proximity, and they will likely share the same ground. It doesn't seem to be an easy problem to solve. Some preliminary results will be more convincing.

Answer 6. As the Reviewer correctly pointed out, the proposed solution could not be directly applied to our device because of the close proximity of EOM and SNSPDs contact pads. However, in Fig. S3c,d of Supplementary Information we show some preliminary results on a double-probe configuration (thus, a solution close to the one initially proposed) mounted on a custom-made holder compatible with our cryostat.

We hope to have answered all questions of the referee satisfactorily. We hope that with the additional experimental data and the revisions to the manuscript she/he will support publication of our results.

Response to Reviewer 3:

Comment 1. *This is a very interesting and potentially important work: implementing high-efficiency and high-speed optical switches and single-photon detectors on the same photonic circuit. If successful, this can have important implications for quantum photonics. The authors have done a very good job developing the device and the work is precisely communicated in the manuscript. My main concern is that the work appears somewhat unfinished therefore not yet exploiting the competitive advantage that the approach may have compared to other methods (nano-mechanical, for instance). While SNSPDs and EOM switches have been implemented before on the LNOI platform, the main novelty is that they are combined. Ideally this would allow very rapid switching on quantum pulses. Unfortunately the authors do only demonstrate explicit modulation at 1 kHz, and high-speed operation is limited by cross-talk in the device. It is argued that this cross-talk can be straightforwardly reduced, however, an explicit demonstration would be a major asset.*

Answer 1. We would like to thank the Reviewer for recognizing the strategic importance of our research work and appreciating the quality of our manuscript. We hope that all the new data we show in this substantially revised version of our paper help in demonstrating the major advantages coming from the monolithic integration of EOMs and SNSPDs on the recently developed LNOI platform.

In this revised version of our manuscript, we have conducted a deeper analysis of electrical crosstalk in our setup and found out that it is less of a concern than what we initially evaluated. Although the noise measured at the readout of the detectors increases linearly with modulation frequency, and reaches a magnitude larger than the SNSPD signal above 100 MHz – making any modulation in this frequency range seemingly impossible – the actual noise current delivered to the SNSPD is much lower. This is due to the kinetic inductance of superconducting nanowires, which naturally counteracts fast variations in their applied current. Thus, upon removing the noise signal with the use of proper frequency filters, a nanowire click can be again visible at the readout of the detector and the normal functionality of the SNSPD is recovered.

Although electrical crosstalk still prevented us to operate the EOM at his full half-wave voltage, we are now able to explicitly show single-photon detection at a modulation frequency up to 1 GHz with a reduced peak-to-peak amplitude of 5 V (around one third of the half-wave voltage). Our results on high-speed modulation measurements are reported in Fig. 5b,c of the new manuscript, and commented in lines 300-331.

We have also complemented our paper with a Supplementary Information material where we analyze in detail the problem of electrical crosstalk, present an electric circuit model which precisely reproduces our experimental observations, and propose viable ways for overcoming this problem in future experiments by backing up our claims with systematic sets of data.

Comment 2. *Furthermore, the SNSPD detector efficiencies are rather low and coupling losses to the device are high. This leaves the impression that there is still a lot of work to be done before the promised quantum-photonics applications can be realized. I would encourage the authors to complete their work and show high-speed switching and efficient single-photon detection on the platform. I think this would be an important result for the community warranting publication in Nat. Comm.*

Answer 2. We thank the referee for pointing this out. Although it is true that single-photon detectors and reconfigurable circuits have been already separately implemented in several photonic platforms, their integration on the same device does not simply imply combining the two components and requires the development of specially tailored multi-steps fabrication processes. For this reason, a fairer comparison of the performance of our device should be made with

research works where these two components have been monolithically integrated on-chip rather than demonstrated independently. In this regard, we would like to note that the first demonstration of a reconfigurable circuit integrated with single-photon detectors has been reported only recently using an opto-mechanical system (S. Gyger et al., Nature Comm. 12, 1408 (2021)). Here, the SNSPDs displayed a system detection efficiency < -30 dB, the employed reconfigurable element required a driving voltage close to 200 V, and their joint operation was shown only at frequencies lower than 1 MHz. Although the performance of our device can be subject to further improvements, we can certainly claim that it already represents a new state-of-the-art reference for photonic systems integrating reconfigurable optical networks and single-photon detectors.

We also would like to stress again the fact that (see Answer5 to Reviewer2) the integration of superconducting nanowires on LNOI waveguides is regarded as a highly challenging task because of the wet etching processes required for the fabrication of low-loss photonic circuits. So far, this step has been successfully accomplished only by two research groups (our work, and Ref. 40 of the new manuscript). Although the SNSPDs demonstrated in Ref. 40 displayed a 46% on-chip detection efficiency, the waveguides suffered from elevated levels of propagation loss (5 dB/cm) because of the employed process for fabrication of superconducting nanowires. Our detectors display a reduced efficiency of around 25%, yet are integrated with waveguides with considerably lower propagation loss (0.2 dB/cm).

Regarding the Reviewer's concern about the fiber-to-chip coupling efficiency: even though the efficiency of our couplers was not extensively optimized, it already stands among one of the best reported to date for gratings implemented on a pure LNOI platform, i.e., without any use of a metal back reflector for an improved directivity, or additional material layers for an enhanced grating strength. In line 91 of the new manuscript we now make explicit reference to one of our recently published works (E. Lomonte et al., Opt. Express 29, 20205-20216 (2021)), where we explain in detail how our grating couplers are designed and how – upon a proper optimization - their coupling efficiency can be boosted up to almost 100% with the aid of a metal back-reflector.

That said, we agree with the Reviewer that the performance of our device should be improved in order to find real applications in quantum photonics. In this regard, we would like to notice that - unlike the very mature Silicon-on-insulator platform - LNOI is still in its infancy and a technology under constant development. We also hope that the Reviewer can understand that the optimization of a new fabrication workflow is a long process subject to the availability of complex cleanroom equipment, and, especially in a University environment, this can require a time frame spanning several months up to more than one year.

Given the potential importance of our work for the quantum photonic community, which is recognized by the same Reviewer, we believe that our results deserve to be published in Nature Communications.

We hope to have answered all questions of the referee satisfactorily. We hope that with the additional experimental data and the revisions to the manuscript she/he will support publication of our results.

Additional modifications, which have not been listed in the point-to-point answer to the reviewers, are the following:

- We included in the abstract our new results on high-speed modulation and DC-bias-drift

Reviewers' Comments:

Reviewer #1:

None

Reviewer #2:

Remarks to the Author:

The authors have taken significant efforts, including additional experiments and analyses, to improve the work. My previous concerns are adequately addressed, and I recommend the revised manuscript for publication.

POINT-BY-POINT RESPONSE

Reviewer #2 (Remarks to the Author):

Comment: The authors have taken significant efforts, including additional experiments and analyses, to improve the work. My previous concerns are adequately addressed, and I recommend the revised manuscript for publication.

Answer: We would like to thank again all the reviewers for their effort in revising our manuscript and for helping us in improving the quality of our research paper. We are delighted to know that they find our revised manuscript suitable for publication.